# Dietary and Nutritional Support in Gastrointestinal Diseases of the Upper Gastrointestinal Tract (I): Esophagus

**DOI:** 10.3390/nu14224819

**Published:** 2022-11-14

**Authors:** Miguel A. Montoro-Huguet

**Affiliations:** 1Unit of Gastroenterology, Hepatology & Nutrition, University Hospital San Jorge, 22005 Huesca, Spain; maimontoro@gmail.com; 2Department of Medicine, Psychiatry and Dermatology, University of Zaragoza, 50009 Zaragoza, Spain; 3Aragón Health Research Institute (IIS Aragón), 50009 Zaragoza, Spain; 4Aragón Health Sciences Institute (IACS), 50009 Zaragoza, Spain

**Keywords:** dysphagia, achalasia, eosinophilic esophagitis, caustics, gastroesophageal reflux, obesity, Barrett’s esophagus, esophageal adenocarcinoma, enteral nutrition, parenteral nutrition

## Abstract

The esophagus is the centerpiece of the digestive system of individuals and plays an essential role in transporting swallowed nutrients to the stomach. Diseases of the esophagus can alter this mechanism either by causing anatomical damage that obstructs the lumen of the organ (e.g., peptic, or eosinophilic stricture) or by generating severe motility disorders that impair the progression of the alimentary bolus (e.g., severe dysphagia of neurological origin or achalasia). In all cases, nutrient assimilation may be compromised. In some cases (e.g., ingestion of corrosive agents), a hypercatabolic state is generated, which increases resting energy expenditure. This manuscript reviews current clinical guidelines on the dietary and nutritional management of esophageal disorders such as severe oropharyngeal dysphagia, achalasia, eosinophilic esophagitis, lesions by caustics, and gastroesophageal reflux disease and its complications (Barrett’s esophagus and adenocarcinoma). The importance of nutritional support in improving outcomes is also highlighted.

## 1. Introduction

The esophagus is a hollow muscular tube of 18 to 26 cm that acts as a conduit for the transport of food from the oral cavity to the stomach [1]. This organ has a sphincter at each end that joins the hypopharynx above to the stomach below. Structurally, the esophageal wall is composed of four layers: innermost mucosa, with an inner “skin-like” lining of stratified squamous epithelium; submucosa; muscularis propria; and outermost adventitia. The neuroanatomical control of esophageal function is highly complex [2]. It involves parasympathetic and sympathetic nerves ultimately responsible for peristalsis and sensation perception, affecting chemoreceptors located in the esophageal mucosa and submucosa and/or mechanoreceptors in the esophageal musculature [1,2,3]. Disruption of these mechanisms due to anatomical or structural damage (macro- or microscopic) or severe dysfunction of esophagus neuromuscular function seriously affects the organ’s ability to perform its purpose [4]. This means that macro- and micronutrients do not reach the stomach, leading to varying degrees of malnutrition and, in the most severe cases, dehydration, electrolyte depletion, and starvation.

This article summarizes the consequences of the most common esophageal diseases on nutritional status and underlines the importance of dietary and nutritional intervention to improve outcomes. Diseases that have been the subject of this review include severe oropharyngeal dysphagia, achalasia, eosinophilic esophagitis (EoE), corrosive agent-induced lesions, and gastroesophageal reflux disease (GERD). All of these diseases cause difficulties in providing the adequate caloric–protein intake required for the metabolic functions of the organism. In addition, some lead to increased metabolic demands due to concomitant inflammation and stress resulting in excess energy expenditure (e.g., caustic esophagitis or cancer). Such diseases should be addressed by a multidisciplinary team comprising gastroenterologists and registered dietitians or nutritionists in advanced hospital nutrition units [5].

## 2. Aim

This article aims to provide an overview of esophageal diseases in which dietary intervention or advanced nutritional support [6,7] are required to improve health outcomes. It covers most of the clinical conditions that can impair the nutritional status of patients due to macroscopic (e.g., peptic, or neoplastic esophageal stricture), microscopic (e.g., EoE), or motility disorders, either at the oropharyngeal (e.g., Parkinson’s or Alzheimer’s disease) or esophageal (e.g., achalasia) level (Table 1). A special section is dedicated to damage produced by the voluntary or unintentional ingestion of corrosive agents. In the case of the latter, there can be severe malnutrition caused by swallowing difficulties arising from dysphagia and pain as well as by the hypercatabolic state associated with sepsis and inflammation [8]. 

## 3. Methods

The recommendations, guidelines, and advice proposed here are not the result of a systematic literature review but of expert judgment based on a review of the literature to advise on best practice. No formal rating of the quality of evidence or strength of recommendation was performed. However, the guidelines and recommendations issued by the American Society for Parenteral and Enteral Nutrition (ASPEN) and the European Society for Nutrition and Metabolism (ESPEN), to which the author belongs, have been considered [9,10]. Likewise, for each of the diseases under analysis (Table 1), searches of specific databases (Google Scholar, Web of Science (WOS), SCOPUS, EMBASE, PubMed (MED-LINE), and Cochrane Central Register of Controlled Trials up to 30 September 2022) have been conducted using the appropriate terms.

## 4. Dietary and Nutritional Support in Esophageal Diseases

This section briefly overviews the set of GI diseases that a registered dietitian or nutritionist should be aware of to improve their competencies and abilities to deliver highly qualified support based on the best available evidence.

### 4.1. Severe Oropharyngeal Dysphagia

Oropharyngeal dysphagia (OD) is a prevalent condition that is recognized by the World Health Organization (WHO) in the *International Classification of Diseases* [11]. Patients with OD have difficulty transferring food from the mouth to the pharynx and report the feeling of an obstruction in the neck. The most severe cases are accompanied by coughing, choking, drooling, and violent nasal regurgitation when attempting to swallow liquids or solids. A history of aspiration pneumonia is common in such cases, and gradual and progressive weight loss is the norm. 

Clinical conditions in which OD may develop include older age, neurodegenerative diseases (Parkinson’s disease, Alzheimer’s, motor neuron disease), previous stroke, traumatic brain injury, and head and neck cancer [11,12,13,14,15,16,17,18,19,20,21,22,23,24,25,26] (Table 2). Approximately 50–75% of patients with OD present impaired safety of swallowing with bolus penetration into the laryngeal vestibule, and 20–25% of these experience aspiration into the airway [11,27,28]. A thorough history and physical examination are essential to identify the underlying etiology, considering iatrogenic, infectious, metabolic, myopathic, neurologic, and structural causes. The diagnostic evaluation includes laboratory tests, imaging tests to exclude brain damage, fiberoptic endoscopic evaluation of swallowing (nasoendoscopy), nasopharyngeal laryngoscopy, videofluoroscopy, and manometry [29,30,31,32,33].

Depending on the cause, some patients with severe OD will respond to medical or surgical treatment, which resolves concomitant nutritional impairment. Such is the case of patients with cricopharyngeal dysfunction, which can in many cases be alleviated by myotomy, endoscopic dilations, or botulinum toxin injection [34,35]. Unfortunately, in patients with oropharyngeal dysphagia following a stroke or severe neuromuscular disease, such as myasthenia gravis, Parkinson’s disease, multiple sclerosis, and amyotrophic lateral sclerosis, etiological treatment may not be expeditious or may simply be ineffective. 

The goal of managing OD in patients is to improve food transfer, prevent aspiration, and ensure adequate supply of the caloric–protein requirements. Treatment includes swallow rehabilitation therapy and nutritional support. In this strategy, it is essential to generate “Units Specialized in the Management of Swallowing Disorders”, where registered dieticians or nutritionists, neurogastroenterologists, geriatricians, experts in endocrinology and nutrition, and skilled nursing staff can collaborate following a multidisciplinary protocol. 

Some dietary interventions may help to improve swallowing and minimize the risk of aspiration:(1)For those patients who do not show adequate tolerance of liquids, the use of certain additives with thickening properties may be helpful in improving their swallowing ability. The European Society for Swallowing Disorders (ESSD) has described the evidence in the literature on the effect that bolus modification has upon the physiology, efficacy, and safety of swallowing in adults with OD of diverse etiologies [11]. These studies show that increasing the viscosity from liquid to nectar and pudding reduces the prevalence of penetrations and aspirations, suggesting that patients with OD do indeed benefit from taking fluids with increased viscosity, which reduces the risk of laryngeal penetration and/or aspiration [11,36,37];(2)The transfer of the food bolus can be improved if the mouthfuls are small in volume;(3)Alternation of solid and liquid boluses can also facilitate transfer;(4)For those patients with severe dysphagia of neurological origin, the assistance of a caregiver may be critical, and the meals should be administered during times of maximal attentiveness;(5)Finally, for those patients who are refractory to all these measures or at high risk for aspiration (e.g., severe neuromuscular dysfunction), enteral nutrition should be provided, preferably by endoscopic, percutaneous gastrostomy. If the patient also has gastroparesis, a double lumen feeding tube can be attempted, whereby one is placed in the stomach to aspirate the gastric remnant (e.g., biliary reflux) and the other in the duodenum for nutrient perfusion.

### 4.2. Achalasia

Achalasia is a relatively rare primary motor esophageal disorder characterized by the ab sence of relaxations of the lower esophageal sphincter (LES) and of peristalsis along the esophageal body. This esophageal motility disorder is thought to result from the progressive degeneration of ganglion cells in the myenteric plexus of the esophageal wall. The gold standard test for diagnosing achalasia is high-resolution manometry (HRM) of the esophagus. HRM should be the next test after first excluding the possibility of mechanical obstruction by endoscopy [38]. The incidence of achalasia ranges from 0.3 to 3.0/100,000 adults, and the prevalence ranges from 1.8 to 12.6/100,000 [38,39]. However, there appears to be striking international variations and significant differences within countries [40] and, in any case, the incidence and prevalence seem to be increasing in very different geographic areas [38,40]. The disease occurs with equal frequency in men and women and is more common with advanced age [41]. Clinically, it manifests as mixed dysphagia (solids and liquids), initially paradoxical and intermittent, and is ultimately progressive, causing marked dilatation of the esophagus (Figure 1). In this phase, regurgitation, sialorrhea, and nocturnal coughing outbreaks are common, reflecting slight bronchoaspiration. If the disorder is not corrected, the dysphagia becomes disabling, and the patient is destined to severe malnutrition with an estimated average weight loss of 20 ± 16 pounds [42]. It is unclear why certain patients lose significantly more weight than others. In the Patel D et al. case series, weight loss was reported in 51/100 (51%) patients. BMI was lower in patients who reported weight loss (25 vs. 31, *p* < 0.001) with a median weight loss of 28 lbs (14–40 lbs). Weight loss was influenced by achalasia phenotype. Thus, more patients (63%) with type II achalasia (defined as an increased median of integrated relaxation pressure (IRP), 100% failed esophageal peristalsis, and the presence of panesophageal pressurization in ≥20% of swallows) reported higher weight loss compared with other subtypes (*p* = 0.013). In total, 73% of type III achalasia (defined as increased median IRP, presence of ≥20% swallows with premature/spastic contraction, and no evidence of peristalsis) denied having weight loss [43,44].

**Table 2 nutrients-14-04819-t002:** Prevalence of oropharyngeal dysphagia in different settings and populations.

Clinical Condition	Percentage	Reference
Older age	15–40%	[39,40,41,42]
Stroke	37–78%	[39,40,43,44]
Neurodegenerative diseases		
○ Parkinson’s disease	52–82%	[39,40,45,46]
○ Alzheimer’s disease	57–84%	[39,40,47,48,49]
○ Motor neuron disease	30–100%	[50]
Traumatic brain injury	25	[40,51,52]
Head and neck cancer	44–50%	[39,53,54]

Treatment includes:Smooth muscle relaxants;Botulinum toxin injections to the lower sphincter;Pneumatic dilation;Heller myotomy;Peroral endoscopic myotomy.

The decision making regarding treatment must be individualized, and the participation of a multidisciplinary team is always advisable [45,46,47,48,55]. While patients are waiting for an expeditious solution to their disease, some dietary measures should be adopted. These basically consist of frequent, small-volume meals with low fiber content and a high liquid content. In truly disabling cases, where oral feeding is not feasible, the patient should be hydrated and fed intravenously until the time of surgery. Although the disease cannot be cured, most patients can return to near-normal swallowing and a regular diet with appropriate therapy [38]. 

### 4.3. Eosinophilic Esophagitis

Eosinophilic esophagitis (EoE) is a chronic, immune-mediated disorder that involves the esophagus. Its pathogenesis appears to depend largely upon delayed, cell-mediated hypersensitivity, and it is one of the most prevalent esophageal diseases and the leading cause of dysphagia and food impaction in children and young adults [49]. The incidence of EoE appears to be increasing, approaching that of inflammatory bowel disease [50,51]. Approximately 70% of children and adults affected by the disease are male, and the average age of presentation in the adult population is 34 years (range 14 to 77 years) [52]. The diagnosis appears more common in urban versus rural settings, widely reported in North and South America, Europe, Asia, and Australia, however, there have been no published reports from Africa. In the United States, EoE is more frequent in cold and arid areas than in areas with a more tropical climate [53].

Diagnosis requires all of the following [54]: (1) symptoms related to esophageal dysfunction; (2) eosinophil-predominant inflammation on esophageal biopsy, characteristically consisting of a peak value of ≥15 eosinophils per high-power field (HPF) (or 60 eosinophils per mm^2^); (3) exclusion of other causes that may be responsible for or contributing to symptoms and esophageal eosinophilia (Figure 2).

EoE is often diagnosed late, with an average time delay of up to 6 years (interquartile range 2 to 12 years) [56,57]. This is likely mainly due to the nonspecific nature of the symptoms in children and adults. Thus, some of the symptoms observed in children under 10 years of age may not be sufficiently clear to raise diagnostic suspicion. These symptoms include food refusal, poor appetite, “picky eating”, trouble with the inclusion of new foods into the diet, preference for softer foods and liquids, and a slow pace of eating [56]. 

Despite the dietary limitations and restrictions to which these patients are subjected, studies that have evaluated nutritional parameters (weight, height, BMI, macronutrients, and vitamins) have not shown evident dietary deficiencies in these patients, with some exceptions. Figure 3 shows some potential causes that could explain states of malnutrition (or growth retardation in children) in patients with EoE [54].

The goal of treatment is to achieve symptom relief, induce histologic remission, and prevent or treat complications (e.g., fibrotic esophageal stricture). The management of EoE includes dietary, pharmacologic, and endoscopic interventions. Dietary intervention, proton pump inhibitors (PPIs), or topical glucocorticoids are first-line options for the initial treatment of EoE (Table 3). The choice of first-line therapy is patient-dependent, and each has its benefits and disadvantages [55,58,59,60,61,62,63,64,65,66,67,68,69,70,71,72,73,74,75]; there are several excellent reviews on this subject [49,58,59,60,61,62,63].

**Dietary therapy** is an effective first-line treatment for EoE in children and adults. The selection of foods to be eliminated, the duration of restriction, and how restricted foods should be reintroduced have been controversial topics over time. We highlight here some issues of practical interest:(1)The patient or their caregivers should be informed about the pros and cons of each available option before planning;(2)When available, a registered dietician or nutritionist who is familiar with food allergies is a very valuable part of the patient’s care team;(3)A diet based on the empirical elimination of six foods (milk, wheat, egg, soy/legumes, nuts, fish/seafood) (6-FED) was initially considered the gold standard for the management of EoE [76,77,78]. The 6-FED followed by an endoscopic procedure before reintroducing a food and after checking histological healing showed that cow milk (especially in children < 10 years old), wheat, egg, legumes and, to a lesser extent, soy were the most common food triggers for EoE in both children and adults [78]. Nevertheless, this dietary approach did not become popular for patients and caregivers due to the need for numerous endoscopies and the high level of restriction for almost a year [76];(4)The most commonly used empiric elimination diet in our patients is the 4-FED, which involves the elimination of cow milk, hen eggs, soy–legumes, and wheat. This approach achieves histologic remissions in up to 54% and 64% of adults and children, respectively [79,80];(5)Studies of 4-FED demonstrated that around half of responders had one or two food triggers (usually milk and wheat) [79]. Following these findings, some authors advocate starting with a two-food (cow milk and wheat) elimination diet, and in the case of nonresponse, sequentially escalating the diet to 4-FED and then 6-FED. This strategy achieves 43%, 60%, and 75% histologic remission rates, respectively [81]. Therefore, an empirical staged elimination diet, starting with one or two food groups, represents a pragmatic dietary approach for children and adult patients with EoE [58];(6)Of interest, Spanish authors have investigated the tolerance of sterilized cow milk (boiled instead of UHT-processing) regarding maintenance of EoE remission, health-related quality of life (HRQoL), nutritional intake, and allergic sensitization in patients of all ages with milk-triggered EoE. Notably, the results of this elegant study demonstrate that sterilized milk did not trigger EoE in two-thirds of patients with documented milk-induced EoE in either the short or long term [82];(7)The highest success rates in symptomatic and histologic improvement are seen with the elemental diet. However, even in children, this diet is the most difficult to follow. This diet should be restricted to patients who have not responded to any other approach, when seeking a nutrient supply that cannot be achieved by any other means, or the patient manifests a desire to initiate treatment in this way and if resources allow it.

Figure 4 shows some practical advice on the timing of reintroducing food. This algorithm is based on following principle: 

“*It is always important to take into consideration that patient reports of positive and negative symptoms have been repeatedly shown not to correlate with endoscopic or histologic evidence of disease remission*” [80,81].

### 4.4. Caustic Injuries

Caustic ingestion continues to be a severe problem, often with devastating consequences, and involves children and adults affected by alcoholism, mental disorders, or suicidal intention (Figure 5). The most affected parts are the oropharyngeal cavity, larynx, esophagus, and stomach [83]. The severity and extent of esophageal and gastric damage are determined by the ingested substance’s corrosive properties (pH), concentration and amount ingested (in adults, a normal sip is 30–50 mL, a large gulp is 60–90 mL), physical form of the agent (solid or liquid), and duration of contact with the mucosa [83,84,85,86,87,88,89,90]. Most patients present with mild injuries that recover without sequelae. However, patients who have ingested a significant amount of strong alkali or acid, usually due to a suicide attempt or psychiatric illness, are at obvious risk of developing severe complications, such as perforation, mediastinitis or peritonitis. Other sequelae such as fistula formation (tracheobronchial, aortoenteric), hemorrhage, or pulmonary complications lead to a state of systemic inflammatory response and increase the basal energy expenditure [the amount of energy required to maintain the body’s normal metabolic activity], creating an imbalance between the nutrient intake and the real (increased) needs, thus contributing to malnutrition [91].

**Table 3 nutrients-14-04819-t003:** Pharmacological management of EoE (first-line treatments in adults).

*Therapy*	*Further Information*
**Proton Pump Inhibitors (PPIs)**	
▪ Dose: standard full-dose PPI once daily ^1^ ▪ Assess symptomatic response at 8 weeks ▪ Clinical response: 61% [61] and 71% [62] ▪ Histological remission: 51% [61] and 49% [62] ▪ Responders: continue the PPI at the lowest dose successful at controlling symptoms # ▪ Nonresponders ##	^1^ Increase the dose up to twice daily in the absence of symptomatic relief after 4 weeks. Alternatively, initiate PPI with a twice-daily dose.
**Fluticasone propionate**	
▪ Dose: ○ 880 mcg/24 h (divide the dose twice a day) ^1^ ○ Orodispersable tablet: 1.5–3.0 mg twice daily ^1^ ▪ Therapy is given for four to eight weeks ▪ Clinical response: 64.1% [63] ▪ Histological remission [63]: ○ Complete: 57.8% ○ Partial: 81.12% ▪ Responders: reduce the dose gradually (e.g., to 50% of the initial dose) over a period of 8–12 weeks # ^2^ ▪ Relapse rate after dose withdrawal: 14–91% [64] ▪ Nonresponders ##	^1^ Patients should not eat or drink for 30 min following administration remission. ^2^ For patients with episodic or seasonal flares, fluticasone may be administered on request rather than as daily therapy.
**Budesonide** [55,65,66,67,68,69,70,71,72,73]	
▪ Dose: ○ Oral suspension: 2 mg twice daily ^1^ ○ Orodispersable tablet: 0.5–1.0 mg twice daily ^2^ ▪ Clinical and histologic response #: ○ Oral suspension: 53% (≤6 eos/hpf) ^3^ ○ Orodispersable tablet: 73.5–75% (≤15 eos/hpf and symptom resolution) ^4^ ▪ Responders: reduce the dose gradually (e.g., to 50% of the initial dose) over a period of 8–12 weeks # ▪ Nonresponders ##	^1^ Viscous budesonide can be compounded by mixing two or four 0.5 mg/2 mL Pulmicort Respules with sucralose (Splenda; 10 × 1 g packets per 1 mg of budesonide, creating a volume of approximately 8 mL) [55,65]. ^2^ Limited availability [55,67]. ^3^ After 12 weeks. ^4^ After 48 weeks.
**Mometasone furoate**^1^ Dose:	
○ Swallowed aerosolization: 200 mg 4 times daily ○ Methylcellulose oral suspension: 500–1500 mg once daily	^1^ Has only been tested in children. Because of its lower bioavailability, it has potentially fewer adverse effects than other steroids [85].
**Ciclesonide** ^1^ Dose:	
○ Swallowed aerosolization 160–1280 mg once daily.	^1^ Has only been tested in children. Because of its lower bioavailability, it has potentially fewer adverse effects than other steroids [85]

# After 8–12 weeks, endoscopy should always be applied in patients with a history of food impaction, presence of strictures, or if <18 years old. For the remaining patients, endoscopy should be offered for histological verification of remission. ## Nonresponders: consider increasing the dose or indicate any of the other therapies that are regarded as first-line.

The evaluation of the degree and extent of the injury is a critical point in decision-making related to nutritional support. Thus, CT of the neck, thorax, and abdomen should be per-formed 3–6 h after ingestion to classify the severity of the lesions in a non-invasive manner (Table 4) [88,92]. In turn, endoscopy should be performed in the first 24–48 h, especially when CT is not available, administering of intravenous contrast is contraindicated, or CT shows signs of wall necrosis, but the interpretation of the findings is uncertain (grade 2 CT). Endoscopy performed under these conditions allows grading the lesions from lesser to greater severity (Zargar classification) and has been shown to have prognostic value [89,90]. Both examinations, together with the patient’s clinical condition, allow us to guide the nutritional management according to the following postulates:

(1)In asymptomatic patients without oral burns and a history of low-volume, accidental ingestion of low-concentration acid or alkali, upper endoscopy is not necessary. Such patients may be discharged from the hospital, and a diet based on soft foods or liquids for the first 24–48 h is recommended;(2)Patients who have ingested a substance with a high risk of esophageal injury (high-concentration acid or alkali or a high volume (>200 mL) of a low-concentration acid or alkali) should be hospitalized. Nutritional support should be initiated with hemodynamic stabilization and the restoration of fluids, electrolytes, and acid–base balance [92];(3)Corrosive ingestion injuries up to *Zargar 2A-grade 1-CT* (low-grade injuries) do not cause long-term sequelae and do not require advanced nutrition. Oral feeding should be reintroduced as soon as patients are swallowing normally, and they should be discharged quickly from the hospital (usually within the first 24–48 h) [92,93];(4)Patients with *grade 2A-CT* esophageal injuries have a low risk (<20%) of stricture formation [94]. Oral nutrition is usually well tolerated and should be introduced as soon as pain diminishes, and patients can swallow. Oral liquids are allowed after the first 48 h if the patient is able to swallow saliva. If patients are unable to tolerate oral liquids, early enteral feeding is provided through a nasojejunal tube or jejunostomy;(5)Patients with *grade 2-CT* lesions will develop stricture in 80% of cases. Pain, sialorrhea, and odynophagia are frequent during the acute phase and can severely limit swallowing, making oral feeding impossible. Such cases may benefit from nutritional support by the nasoenteral route, jejunostomy and, as a last resort, exclusive parenteral nutrition. The decision to adopt one procedure over another is dependent on how long before the patient is expected to be able to restart feeding and their tolerance [91,94]. Kochhar et al. compared the nutritional parameters of 53 and 43 patients with severe acute corrosive injury supplied with nasoenteral tube (NETF) or jejunostomy feeding (JF), respectively. NETF was found to be as effective as JF in maintaining nutrition, and the rate of complications was similar (including the development of strictures). However, NETF provided a lumen for dilatation that was useful as a guide for performing the procedure [95];(6)Signs of perforation (e.g., mediastinitis, peritonitis), major metabolic disorders, and CT evidence of transmural necrosis of the esophagus or stomach (grade 3-CT) in patients are indications for emergency surgery. In all these cases, the surgeon disrupts the continuity of the gastrointestinal tract to save the patient’s life, making oral feeding virtually impossible. The most common interventions in this scenario are:
▪Esophagogastrectomy through a combined abdominal cervical approach. After surgery, patients are left with a cervical esophagostomy (spit fistula), a defunctionalized duodenum, and a feeding jejunostomy [96,97]. One-stage reconstruction after emergency esophagectomy is not advisable because the subsequent development of pharyngeal strictures might compromise outcomes [98]. Whenever necrosis in the upper two-thirds of the esophagus is seen, tracheobronchial endoscopy must be performed before surgery for the detection of tracheobronchial necrosis, which would alter the surgical management (e.g., pulmonary patch repair, typically through a right thoracotomy approach) [99];▪If necrosis is confined to the stomach, total gastrectomy with preservation of the native esophagus should be considered. Although immediate esophagojejunostomy reconstruction has been shown to be safe in a high-volume referral center, with leaks in 5–8% of cases [100], other surgeons prefer to leave protective jejunostomy until after definitive reconstruction by means of a retrosternal ileocolonic esophagoplasty 4 to 8 months after the initial operation. In the interim, the patient can receive their nutritional needs through jejunostomy;▪Concomitant necrosis in about 20% of patients undergoing esophagogastrectomy for causative ingestion requires the excision of additional abdominal organs such as the spleen, colon, small bowel, duodenum, or pancreas. In some patients, it is necessary to perform a proximal pancreatoduodenectomy for duodenal or pancreatic necrosis. The main complication related to this procedure is pancreatic fistula, which may be medically treated [100]. The most experienced surgeons seal the main pancreatic duct and avoid pancreatojejunostomy because of the combined presence of soft healthy pancreatic tissue, peritoneal inflammation, and frequent hemodynamic instability in the postoperative period. Such patients may be nourished by temporarily employing a jejunostomy [93,100];▪Resection should be abandoned if extensive bowel necrosis is found upon laparotomy because of poor survival and issues of compromised nutrition [98,99];▪In summary, the construction of a feeding jejunostomy at the end of surgery (irrespective of the conducted procedure) enables early enteral nutrition in patients with compromised digestive function [93].(7)Esophageal strictures are the most common complication of caustic esophageal ingestion and can affect the esophagus, stomach, and other locations in the digestive tract. They usually develop within 2 months (3 weeks to 1 year) and multiple strictures appear in some cases [98]. Again, nutritional support plays a role in management. Exclusive enteral nutrition is indicated in the following contexts: ▪In patients with esophageal strictures, when endoscopic dilatation is complicated by perforation (4−17%). Esophageal perforations in this context are usually contained and can benefit from non-operative management [98]. In such cases, the nasoenteral route or jejunostomy can be used depending on the patient’s clinical condition;▪Patients with multiple failed attempts at endoscopic dilatations should be considered for reconstructive surgery, usually by elective esophageal resection with esophagogastric anastomosis or colonic interposition;▪Patients with pharyngoesophageal strictures (0.7–6%) that require retrograde or anterograde dilation and/or surgical reconstruction with colonic interposition and/or myocutaneous flap inlay;▪Patients with gastric strictures (75–89% located in the antrum) and who present a perforation complicating the outcome of endoscopic dilatation (3.4–46%) or after stent implantation [101,102]. Many surgeons prefer to perform resection or bypass, which are associated with very low morbidity and mortality rates [103]. 

**Figure 4 nutrients-14-04819-f004:**
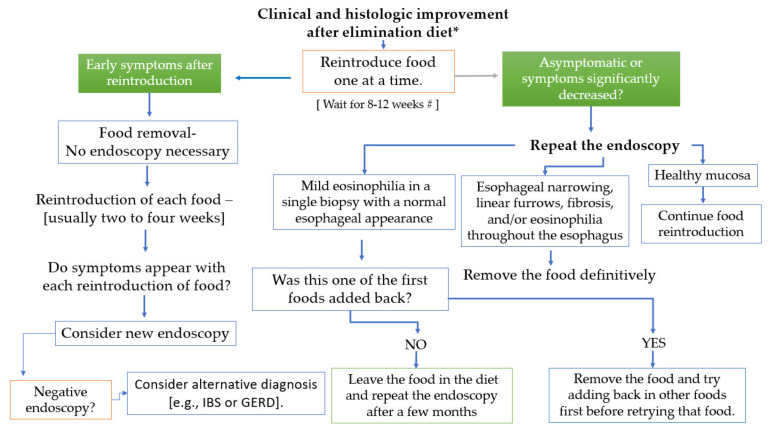
Food reintroduction protocol after verifying symptomatic and histological remission eight weeks after initiating a dietary intervention. * British Society of Gastroenterology (BSG) and British Society of Paediatric Gastroenterology, Hepatology and Nutrition (BSPGHAN) recommend starting with the 2-food elimination diet Ref. [94]. However, the algorithm also applies to 4- or 6-FEDs. [#] two to four weeks is a reasonable alternative clinical approach. [#] two to four weeks is a reasonable alternative clinical approach. IBS: irritable bowel syndrome; GERD: gastroesophageal reflux disease.

**Figure 5 nutrients-14-04819-f005:**
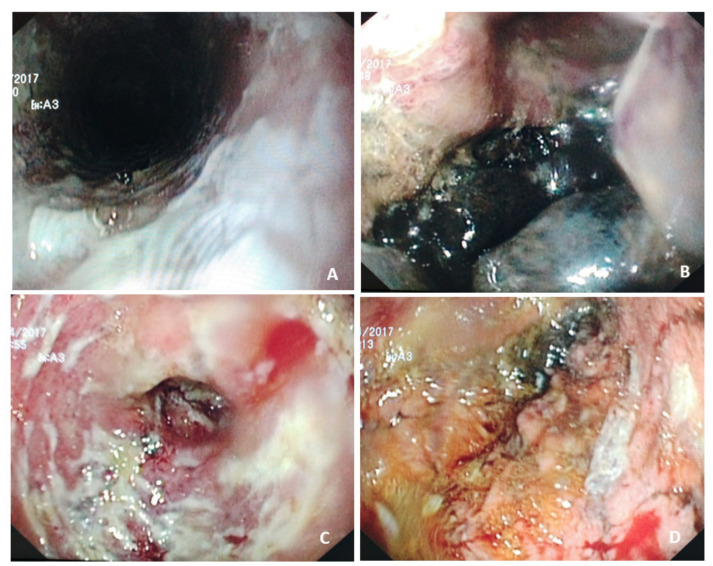
High-grade injuries caused by caustic ingestion. The images correspond to an 82-year-old woman who ingested hydrochloric acid for suicidal purposes due to severe pain associated with polyarthritis and significant dependence for activities of daily living. (**A**) Esophagus. From the proximal esophagus, superficial ulcers can be seen extending around the entire circumference (Zargar IIb). (**B**) Gastric fundus. Remains of gastric contents are observed on an unstructured and friable mucosa, scattered ulcers, and extensive necrosis in the distal portion of the greater curvature. (**C**)Antrum and pylorus. Edematous and friable antral mucosa with deep ulcers with a fibrinous base. (**D**) Gastric necrosis. The area of mucosa corresponds to a segment of approximately 4 cm in the greater curvature of necrotic appearance (Zargar IIIb). This case highlights the pathogenic mechanisms of corrosive lesions induced by strong acids. Typically, as the acid flows along the lesser curvature of the stomach toward the pylorus, pylorospasm impairs emptying into the duodenum, producing stagnation and injury that is more prominent in the antrum. Courtesy of Sánchez Alonso M.D. and Olivencia Palomar M.D., Hospital General Universitario de Ciudad Real (Spain).

### 4.5. Gastroesophageal Reflux Disease

The passage of gastric contents into the esophagus (gastroesophageal reflux [GR]) is a normal physiologic process. Physiologic reflux episodes typically occur postprandially, are short-lived, asymptomatic, and rarely occur during sleep. The term gastroesophageal reflux disease (GERD) is applied to patients with bothersome symptoms suggestive of reflux with or without oesophagitis. 

GERD is a common clinical problem [104]. In fact, GR was the most frequent outpatient diagnosis, with almost 9 million visits, in the United States of America in 2009 [104]. According to US population surveys, 44% of all North Americans experience heartburn at least once per month, 14% at least once per week, and up to 7% daily [105,106]. Similar trends have been observed in other geographic areas [107,108]. GERD is more prevalent in urban residents, women, and older and obese individuals [105,106,107,108]. Clinically, it is characterized by symptoms of substernal burning or acid regurgitation produced by the abnormal GR. Other symptoms include chest pain, chronic and unexplained cough, dysphonia or hoarseness, throat clearing, trouble swallowing (especially if a peptic esophagitis is present), and nausea or vomiting. A subset of patients will develop complications such as peptic stricture, Barrett’s esophagus (BE), and cancer, which represent a significant burden on healthcare systems (Figure 6) [104].

#### 4.5.1. Foods That Contribute to the Triggering or Worsening of Symptoms

In the etiopathogenesis of GERD, there is an imbalance between aggressive factors (number and duration of reflux episodes and acidity of the refluxed material) and defensive factors (esophageal acid clearance and mucosal integrity). Without a doubt, the most important pathogenic factor is incompetence of the lower esophageal sphincter (LES). The three dominant pathophysiologic mechanisms causing esophagogastric junction incompetence are transient LES relaxations, a hypotensive LES, and anatomic disruption of the gastroesophageal junction, often associated with a hiatal hernia (Figure 6). Knowledge of these mechanisms is essential as it has been postulated that some foods that stimulate gastric acid secretion (e.g., caffeine, alcohol) or decrease LES tone (e.g., fatty meals, cocoa, chocolates, and alcohol) should be significantly reduced in the usual dietary intake of a person with GERD [109,110,111,112,113,114,115,116,117,118,119,120,121,122,123,124,125,126,127,128,129,130]. 

In an evidence-based approach, Kaltenbach et al. researched the efficacy of lifestyle measures in GERD management. Certainly, although there was physiologic evidence that exposure to tobacco, alcohol, chocolate, and high-fat meals decreases LES pressure, the results of the review did not provide tangible evidence that intervention on these factors was associated with improvement regarding esophageal pH profiles or symptoms (evidence B: cohort or case–control trials, nonrandomized or uncontrolled clinical trials) [109]. It should be noted that other alternatives are available for controlling symptoms associated with gastric acid secretion (e.g., antisecretory drugs, antacids, and alginates). The authors concluded that larger prospective controlled trials are warranted before dietary and lifestyle modifications can be conclusively recommended in the treatment of GERD [110,111]. Nevertheless, the clinician should keep in mind that eliminating tobacco, high-grade alcohol, and foods rich in sugars and saturated fats is part of an overall strategy to improve individuals’ overall health, and that the presence of reflux symptoms constitutes an excellent opportunity to dissuade patients from the abusive consumption of these “*dietary triggers*”.

#### 4.5.2. Overweight and Obesity

Epidemiological studies show that obesity is a risk factor for GERD because of multiple factors, including increased (1) intra-abdominal pressure, (2) gastroesophageal sphincter gradient, and (3) incidence of hiatal hernia as well as impaired gastric emptying and the output of bile and pancreatic enzymes [109,112,113,114]. Overweight/obesity increases the risk for GERD symptoms by 1.2–3-fold, and 60% of the overweight or obese population report having GERD [115]. In addition, there is evidence that obesity, especially central obesity, increases the risk of complications such as peptic esophagitis, BE, and adenocarcinoma of the esophagus [116,117,118]. At least two systematic reviews [109,111], the first of which included 16 clinical studies [109], have demonstrated that weight loss and bed elevation, among different examined lifestyle interventions, were effective for the resolution of GERD symptoms. 

The European Union of Gastroenterology [116] has formulated specific recommendations that highlight the importance of implementing strategies to treat obesity to improve GERD treatment outcomes (Table 5) [119,120,121,122,123,124,125,126,127,128,129,130,131]. These measures include a careful nutritional assessment by means of anthropometric measures (body weight, body height, BMI, waist circumference), dietary regimen, and increased physical activity. In extreme cases of morbid obesity (BMI > 40 kg/m^2^, or >35 kg/m^2^ when there are obesity-related comorbidities), bariatric surgery effectively reduces reflux symptoms and can contribute to regression in some cases of Barrett’s esophagus [129,130,131]. Roux-en-Y gastric bypass is the most effective surgical modality that is associated with weight reduction and the improvement of GERD symptoms, and it can be performed laparoscopically [130].

#### 4.5.3. Food and Reflux Symptoms during Sleep

Postprandial reflux is common in patients with GERD, and patients frequently present with nocturnal symptoms or wake up in the morning with symptoms suggestive of peptic laryngitis. For this reason, it is very common for physicians to advise their patients not to go to bed shortly after dinner and to wait at least 2–3 h after their last meal [109,132,133,134,135]. 

Initial studies evaluating the effects of timing of the evening meal on 24-h intragastric acidity in healthy volunteers [132,133] and GERD patients [134,135] showed different effects on nocturnal pH [109]. Subsequently, a comprehensive review based on meta-analyses, systematic reviews, randomized clinical trials (RCTs), and prospective observational studies [136,137] suggested that avoiding late evening meals and head-of-the-bed elevation is effective against nocturnal GERD. Stanciu et al. demonstrated that the percentage of time during which esophageal pH was below 5 and the number of reflux episodes were significantly reduced when patients were in a bed-up position (elevation of the head end of the bed with blocks of 28 cm) compared with when sitting or lying [138]. New evidence indicates the following:○Elevation using a foam wedge causes a statistically significant decrease in the time that distal esophageal pH is less than 4 compared with the flat position. [139,140];○Elevating the head of the bed is an easy and effective way to alleviate the symptoms of acid regurgitation. Furthermore, this intervention results in more effective relief of symptoms than taking medications alone [141,142];○Elevation using a foam wedge causes a statistically significant decrease in the time that distal esophageal pH is less than 4 compared with the flat position. [139,140];○Elevating the head of the bed is an easy and effective way to alleviate the symptoms of acid regurgitation. Furthermore, this intervention results in more effective relief of symptoms than taking medications alone [141,142];○Elevating the head of the bed may be useful for relieving acid regurgitation among esophageal cancer patients after surgery [141];○The use of a wedge-shaped pillow (WSP) alleviates reflux symptoms in patients with esophageal cancer following esophagectomy and reconstruction. Likewise, the combined treatment (antisecretory drugs + WSP) also reduces the severity of esophagitis [143];○Several studies show that sleeping with the head of the bed elevated or on a wedge reduces GER and lying left-side down reduces GER versus lying right-side down and supine [144]. The left lateral position is a suitable alternative to prone for the postural management of infants with symptomatic GER [145];○Finally, bed head elevation by reducing the time of acid exposure also alleviates the consequences of nocturnal supraesophageal reflux, including perennial nasopharyngitis, cough, and asthma [146].

#### 4.5.4. Dietary, Barrett’s Esophagus, and Cancer Risk

About 10–15% of patients with gastroesophageal reflux disease develop BE. This is considered a premalignant condition because it can progress from metaplasia to high-grade dysplasia and eventually to adenocarcinoma [147]. The incidence of esophageal adenocarcinoma and its precursor, BE, have increased greatly (~500%) over the past 40 years and continue to rise [148,149,150]. Although the reasons are not clear, advanced age, male gender, obesity, smoking, and alcohol have been identified as risk factors [151,152,153,154,155,156,157]. There is an abundance of studies investigating the relationship between obesity and BE and adenocarcinoma, and authors suggest that this relationship may be related to central adiposity. Hence, increased cell turnover and eventual carcinogenesis are likely precipitated by increased intragastric pressure but are also affected by the complex interaction of increased insulin resistance in patients with increased fat mass [153]. Duggan et al. evaluated the association between markers of obesity and progression from BE to esophageal adenocarcinoma in 392 patients enrolled in the Seattle Barrett’s Esophagus Study [152]. Authors found that among patients with BE, increased levels of leptin and insulin resistance were associated with increased risk of esophageal adenocarcinoma. 

Arcidiacono et al. investigated the effect of a 24-month moderate calorie and protein restriction program on overweight or obese patients affected by BE [158]. The improvement in metabolic condition resulted in a downregulation of the extracellular signal-regulated kinase-mediated mitogenic signal in 43.5% of patients, probably affecting the molecular mechanism driving adenocarcinoma development in BE lesions [158]. On the other hand, there is also experimental evidence that a diet rich in fat and refined sugars containing high fructose concentrations alters the gut microbiota and favors esophageal adenocarcinoma carcinogenesis [159]. Other epidemiological studies on dietary intervention suggest that reducing the consumption of a diet rich in fat and refined sugars may exert a favorable effect by decreasing the risk of progression of BE to adenocarcinoma [150,160,161,162].

All this evidence has led some authors to investigate the effectiveness of lifestyle interventions on BE risk. Zhao et al. researched the effects of seven lifestyle factors: smoking, alcohol, BMI, physical activity, sleep time, medication, and diet. They observed statistically significant increased BE risks for smoking, alcohol intake, body fatness, and lower sleep time. Reduced risks of BE were found for aspirin, the use of proton pump inhibitors, and intake of vitamin C, folate, and fiber [162]. This large meta-analysis revealed that lifestyle modifications could reduce the risks of BE and, consequently, esophageal adenocarcinoma. In another study, Wang et al. investigated the potential effects of diet on risk of BE in over 20,000 participants in the Melbourne Collaborative Cohort Study. Positive associations were observed for discretionary food and total fat intake. By contrast, intakes of leafy vegetables and fruit were inversely associated with risk of BE, as were dietary fiber and carotenoids [160]. The results of these studies underscore the concept that public health and clinical guidelines that incorporate dietary recommendations could reduce the risk of BE and, thereby, esophageal adenocarcinoma [160,161,162,163,164,165,166,167,168]. 

#### 4.5.5. Complications Contributing to Malnutrition

Esophageal reflux may result in several complications, including esophagitis, upper gastrointestinal bleeding, anemia, peptic ulcer, peptic stricture, dysphagia, cancer of gastric cardia, and Barrett´s esophagus. All of these can adversely affect nutritional status via various mechanisms (Figure 7).

## 5. Conclusions

Anatomical or swallowing disorders of the esophagus adversely affect nutritional status. Severe dysphagia of neurological origin and sequelae following the ingestion of corrosive agents can have devastating repercussions and require advanced nutritional support. In other cases, it is the food itself that negatively affects esophageal function (e.g., EoE or GERD). Finally, lifestyle and Western dietary patterns generate obesity. Experimental and epidemiological evidence supports the concept that obesity and other harmful habits, such as smoking or alcoholism, increase the risk of BE and esophageal adenocarcinoma. Registered dietitians or nutritionists should have a comprehensive knowledge of all these mechanisms and work interdisciplinarily with gastroenterologists to correct any macro- or micronutrient deficiencies. In some cases, the selective exclusion of foods will be necessary (e.g., EoE) though without the deterioration of nutritional status. This is a challenge for professionals with expertise in human nutrition and dietetics.

## Figures and Tables

**Figure 1 nutrients-14-04819-f001:**
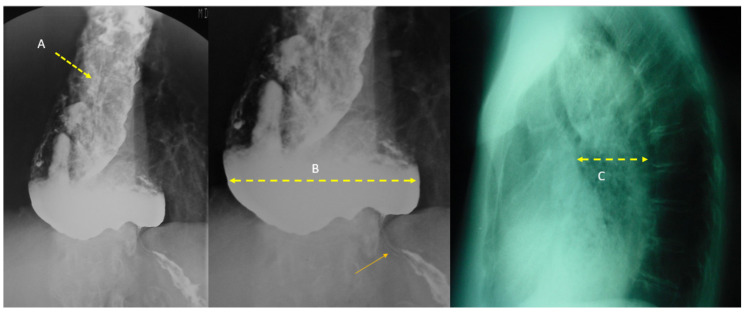
A 52-year-old male presented with marked signs of dehydration and malnutrition after a 20-year history of dysphagia, initially for liquids and later for solids. To overcome difficulty in swallowing, the patient used various maneuvers such as stretching the neck, raising the arms, taking a deep breath, and/or drinking water. In the weeks before his admission, he indicated experiencing excess coughing at night, with regurgitated food remnants found on the pillow the following day. The patient was diagnosed with achalasia and underwent Heller myotomy after prehabilitation with fluid and electrolyte replacement and parenteral nutrition. (**A**) Barium esophagogram with repletion defects corresponding to food detritus and saliva indicated (yellow arrow), (**B**) Marked increase in esophageal diameter (yellow arrow) that more closely resembles a colon (yellow arrow) in barium esophagogram. The sharp end is reminiscent of a mouse tail (orange arrow). (**C**) Chest X-ray showing widening of the mediastinum simulating false cardiomegaly in the lateral projection (yellow arrow. Courtesy of Dr. Rubio, M.D. Department of Radiology University Hospital, San Jorge, Huesca (Aragón-Spain).

**Figure 2 nutrients-14-04819-f002:**
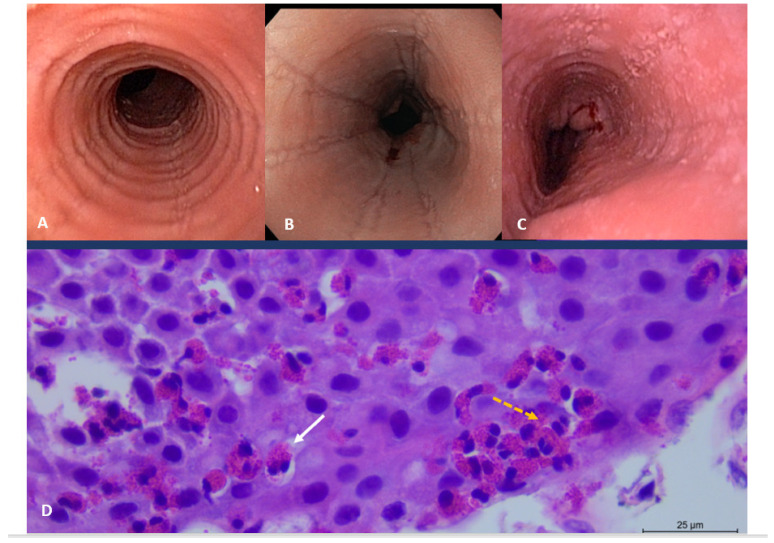
Eosinophilic esophagitis. The figures show the endoscopic appearance and typical histologic features of the disease. (**A**). Circular rings simulating the appearance of a trachea; (**B**). Linear narrows, (**C**). Erosions can be seen in the esophagus. (**D**). H&E esophageal mucosa showing numerous intraepithelial eosinophils (>15 per high-power field) (orange dashed arrow) with eosinophilic microabscesses (white solid arrow). Courtesy of Santos Santolaria, M.D. Ph.D. Unity of Gastrointestinal Endoscopy, University Hospital San Jorge, Carmen Bernal M.D. Ph.D. and Marigil MA M.D., Ph.D., Department of Pathology, University Hospital San Jorge.

**Figure 3 nutrients-14-04819-f003:**
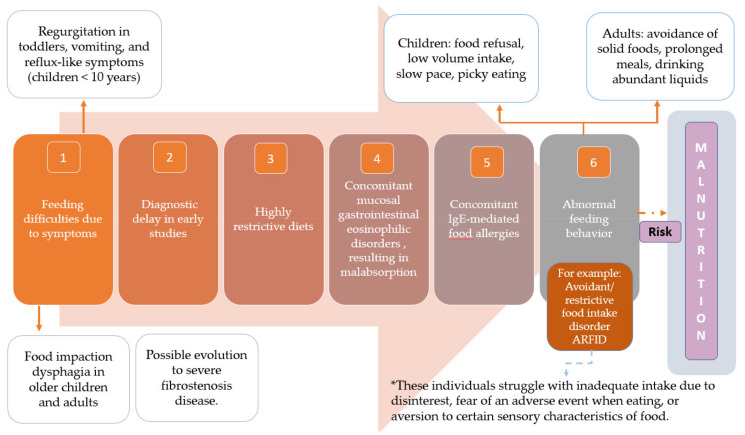
Six potential reasons contributing to the development of malnutrition in children or adults with EoE. Adapted from Molina Infante J with permission (Ref. [57]).

**Figure 6 nutrients-14-04819-f006:**
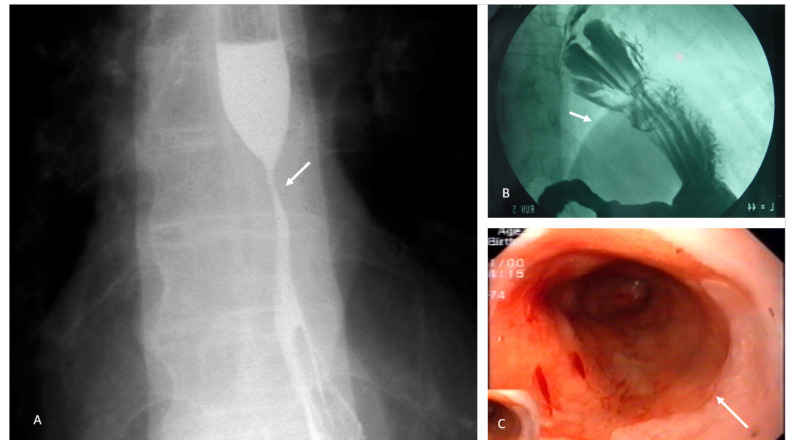
Images related to gastroesophageal reflux disease. (**A**) Filiform narrowing of the lumen of the middle one-third of the esophagus can be seen in a patient with long-standing gastroesophageal reflux symptoms. The stricture results from the reparative collagenization of peptic lesions caused by acid reflux. (**B**) Voluminous hiatal hernia. Note how a considerable proportion of the stomach extends beyond the diaphragm line (arrow). (**C**) Extensive area of salmon-colored mucosa that contrasts with the lighter coloration of the remaining esophagus, suggesting Barrett’s metaplasia. Note the transition between normal mucosa and intestinal metaplasia, and its length (long Barrett’s esophagus) (arrow).

**Figure 7 nutrients-14-04819-f007:**
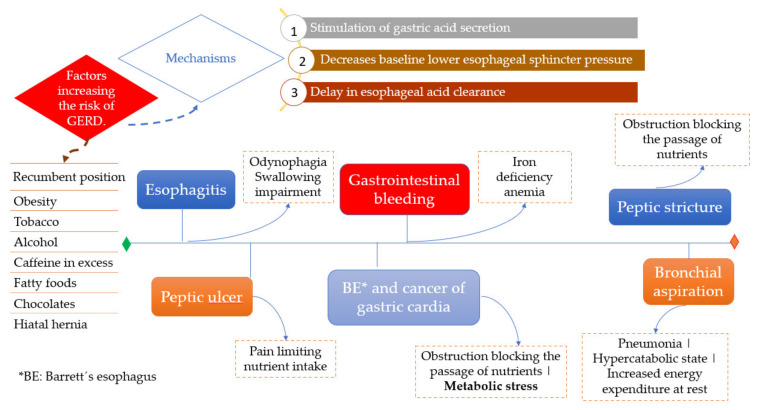
Precipitating factors of GERD, complications, and effects on nutritional status.

**Table 1 nutrients-14-04819-t001:** Diseases of the esophagus may require dietary intervention or specific nutritional support.

*Diseases Caused Principally by Anatomical or Structural Damage*	*Diseases Caused Primordially by Alterations in Neuromuscular Control of Esophagus’ Function*
▪ Extrinsic compression of the oropharynx (e.g., goitre, cervical osteophytes, adenopathies) ▪ Intrinsic lesions of the oropharynx (e.g., radiation therapy, carcinoma) ▪ Extrinsic esophageal compression (e.g., intrathoracic goitre, lymphoma, aberrant subclavian artery–dysphagia lusoria, tuberculosis, left atrial enlargement due to mitral insufficiency) Intrinsic lesions that obstruct the esophageal lumen (e.g., strictures of inflammatory—peptic, actinic, or infectious—or neoplastic origin) ▪ Eosinophilic esophagitis# ▪ Caustic injuries # EoE may also cause secondary motility disorders	Severe oropharyngeal dysphagia of neuromuscular origin (e.g., strokes, Alzheimer’s disease, Parkinson’s disease, multiple sclerosis, amyotrophic lateral sclerosis, myotonic dystrophy, cricopharyngeal achalasia) Severe and prolonged esophageal motility disorders (e.g., achalasia, scleroderma)

**Table 4 nutrients-14-04819-t004:** Classification of the severity of lesions according to computed tomography (CT) [88,92].

	Findings Found on Computed Tomography.
**Grade 1**	Homogenous enhancement of the esophageal wall while wall edema andmediastinal fat stranding are absent ^1^.
**Grade 2a**	Injuries display internal enhancement of the esophageal mucosa and hypodense aspect of the esophageal wall, which appears thickened while concomitant enhancement of the outer esophageal wall may sometimes confer a “target” aspect ^2^.
**Grade 2b**	Injuries present as a fine rim of external wall enhancement: the necrotic mucosa is not enhanced and fills the esophageal lumen, which indicates liquid density. Mediastinal fat stranding is uniformly present in grade 2 esophageal injuries.
**Grade 3**	Transmural necrosis as shown by the absence of post-contrast wall enhancement ^3^.

*Correspondence with Zargar’s Classification.*^1^ Usually corresponds to low grade 0 to 2a endoscopic burns. 0: normal. 1: Edema and hyperemia of the mucosa. 2a: Superficial localised ulcerations, friability, and blisters. ^2^ Corresponds to more severe endoscopic burns. 2b: Circumferential and deep ulcerations. ^3^ Usually corresponds to grade 3b necrosis on endoscopy. 3a: Multiple and deep ulcerations and small scattered areas of necrosis. 3b: Extensive necrosis.

**Table 5 nutrients-14-04819-t005:** Recommendations formulated by the European Union of Gastroenterology for patients with GERD and obesity [116].

Recommendation	Consensus	References
**Screening and Assessment**		
*Recommendation no. 29* Nutritional status screening should be performed for patients with GERD and overweight or obesity, encompassing basic anthropometric measurements (body weight, body height, BMI, waist circumference). *Recommendation 30* Sarcopenia and sarcopenic obesity should be assessed, if there are indicators for sarcopenia, body composition analysis (DXA # or BIA *) and dynamometry (handgrip strength) should be used in GERD patients with overweight or obesity. # DXA: dual-energy X-ray absorptiometry; * BIA: bioelectrical impedance analysis	Grade of recommendation GPP—strong consensus 96% agreement Grade of recommendation GPP—strong consensus 93% agreement	[112,113,114,115,116] [117]
**Treatment**		
*Recommendation 31* Patients with GERD and obesity should be encouraged to lose body weight and reduce waist circumference. *Recommendation 32* Patients with overweight or obesity and GERD should undergo weight reduction preferentially through lifestyle modification including dietary regimen and increased physical activity. *Recommendation 33* In patients with GERD and BMI >40 kg/m^2^, or >35 kg/m^2^ when there are obesity-related comorbidities, bariatric surgery can be considered to achieve weight reduction if nonsurgical interventions failed to achieve the goals. The preferred procedure is Roux-en-Y gastric bypass (RYGB)	Grade of recommendation A—strong consensus 100% agreement Grade of recommendation B—strong consensus 100% agreement Grade of recommendation 0—strong consensus 93% agreement	[118,119,120,121,122,123,124] [109,111,116,125,126,127,128] [129,130,131]

## Data Availability

Not applicable.

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
