# Peer review of "Dietary and Nutritional Support in Gastrointestinal Diseases of the Upper Gastrointestinal Tract (I): Esophagus"

_nutrients, 2022, doi:10.3390/nu14224819_

Round 1

Reviewer 1 Report

Lengthy albeit thorough review of a wide range of oesophageal disorders.  The review gives wordy review and description of the overall definitions, pathogenesis, diagnostic workup and management of the multiple conditions described and the focus of the review ie nutritional complications and their management gets rather lost in the text.  Therefore suggest refocussing on summarising description aetiology and management much more succinctly and focus on the nutritional aspects as implied in the title of the review.  As an example EOE the current state of the art approach to management of EOE is addressed expansively rather than succinctly summarised and nutritional aspects then highlighted. The same criticism in the review also applies to extensive overall review of Caustic ingestion and GERD for example -  Tables all comprehensively discussed but too wordy and too much detail.

Some edits suggested below

Table 1 line (#) 74 “primordially” suggest principally or primarily

“bocio” Spanish suggest goitre in table 1

#87 too wordy suggest edit: Clinical conditions in which OD may develop include older age, ….

#141 “and, 144 in any case, the incidence appears to be increasing. “ Not sure if this is referring worldwide data or regional or country specific…can this be clarified?

#147 evidence for increasing incidence.  Is this in USA, Australia, Europe or worldwide?

#195 “The diagnosis also appears more common in urban as opposed to rural settings, and unlike what was reported in several countries such as North and South America, Europe, Asia, and Australia, there have been no published reports from countries in Africa.”  Suggest edit: The diagnosis appears more common in urban versus rural settings, widely reported in North and South America, Europe, Asia, and Australia, however there have been no published reports from Africa.

#198 ” On the other 198 hand, in the United States, EoE is more frequent in cold and arid areas than in areas with a more tropical climate [53].” Suggest edit: In the United States, EoE is more frequent in cold and arid areas than in areas with a more tropical climate [53].

Table 3 “furoato” Should read furoate

Fig 4: Configuration of the algorithm in the pdf sent to this reviewer was pushed to one side and could not be fully reviewed - needs reconfiguration/adjusting

#302 “usually because of autolytic ideas” edit needed this does not make sense ??suicidal attempt??? not sure clarify with edit

#305 “aorto-enteric), hemorrhage” nutritional consequence would be the least of the patients concern aorto-enteric fistula usually catastrophic and often leads to death. Improve this sentence it does not read well.

Table 4 I think too  much information provided here. Addresses the emergency management of caustic injuries in too much detail for this review of nutritional consequences of oesophageal injury It is far too large and wordy a table and should be condensed. The reader is not directed easily to nutritional aspect of immediate and long term management in this large and wordy table.  Has this table been copied from another review?

#368 in many instances text can be simplified through the paper eg instead of “caloric protein needs” suggest edit: nutrition

Author Response

Dear reviewer, thank you very much for your review. I hope that the changes made have improved the original version. 

Reviewer 2 Report

The manuscript entitled „Dietary and nutritional support in Gastrointestinal Diseases of the Upper gastrointestinal tract (I): esophagus” is very interesting. the work is interesting and raises an important issue, which is nutritional support in diseases of the esophagus, it is a good collection and summary of literature data.

Overall, I think the paper fits perfectly with the theme of Nutrients journal, however, in my opinion some corrections would increase the readability of the work.

Above all, the manuscript should contain a list of abbreviations. There are a lot of them in the text, so such a list would definitely make it easier for the reader to follow the paper.

In my opinion, the figures consisting from several pictures should be better described. Better than using directions like “upper, left, middle, right etc.” would be describing the pictures with letters like it has been provided for figure 5.  It is easier to follow for reader.

Figure 5 is bad quality, If its possible, authors should give better pictures.

Table 3 with its very long description is too chaotic. This part should be rewritten. For example: These informations should be presented as written text and then it can be summarized in table.

Author Response

Dear reviewer, thank you very much for your review. I hope that the changes made have improved the original version. 

Miguel Montoro-Huguet M.D. Ph.D.

Reviewer 3 Report

Huguet in his manuscript emphasized on the importance of lifestyle and nutrition on the health and the disorders related to esophagus. The article also emphasized on the influence of micro and macronutrients effect on the functionality of the organ. The strongest part of the current manuscript is that, it comprises the detailed clinical guidelines on dietary and nutritional managements of esophagus disorders and the complexities associated with them.

Minor comments:

Figure 4 is not formatted and should be corrected.

The author is recommended to undergo a thorough check of the manuscript for typographical and grammatical errors.

Author Response

(The authors gave the same response as above.)
